# Effect of Temperature and Material Flow Gradients on Mechanical Performances of Friction Stir Welded AA6082-T6 Joints

**DOI:** 10.3390/ma15196579

**Published:** 2022-09-22

**Authors:** Xiaotian Ma, Shuangming Xu, Feifan Wang, Yaobang Zhao, Xiangchen Meng, Yuming Xie, Long Wan, Yongxian Huang

**Affiliations:** 1State Key Laboratory of Advanced Welding and Joining, Harbin Institute of Technology, Harbin 150001, China; 2Zhengzhou Research Institute, Harbin Institute of Technology, Zhengzhou 450046, China; 3China Aerospace Science and Technology Corporation, Beijing 100048, China; 4China Academy of Launch Vehicle Technology, Beijing Institute of Astronautical Systems Engineering, Beijing 100076, China; 5Shanghai Spaceflight Precision Machinery Institute, Shanghai 201600, China

**Keywords:** friction stir welding, aluminum alloys, gradient, material flow, microstructure, mechanical properties

## Abstract

The temperature and material flow gradients along the thick section of the weld seriously affect the welding efficiency of friction stir welding in medium-thick plates. Here, the effects of different gradients obtained by the two pins on the weld formation, microstructure, and mechanical properties were compared. The results indicated that the large-tip pin increases heat input and material flow at the bottom, reducing the gradient along the thickness. The large-tip pin increases the welding speed of defect-free joints from 100 mm/min to 500 mm/min compared to the small-tip pin. The ultimate tensile strength and elongation of the joint reached 247 MPa and 8.7%, equal to 80% and 65% of the base metal, respectively. Therefore, reducing the temperature and material flow gradients along the thickness by designing the pin structure is proved to be the key to improving the welding efficiency for thick plates.

## 1. Introduction

Aluminum alloys are primarily employed in aerospace and transportation industries due to their lightweight, high specific strength, and corrosion resistance. With more attention to lightweight and high performance, welded components made of thick plates have been widely employed [1]. Friction stir welding (FSW), as an innovative solid-state joining technique, has been proven to be a successful joining process due to its lower heat input [2,3], which is advantageous for joining medium-thick aluminum alloys.

During the FSW process, the frictional heat generated by inserting a high-speed rotating tool into plates is used to plasticize materials. The insertion of the shoulder inevitably causes weld thinning, which is one of the inherent issues of FSW. It decreases the effective load-bearing area and causes stress concentration on the sharp weld edge, reducing the mechanical properties of the joint [4]. In addition, the shoulder generates the major heat, while the pin is the primary source of material deformation and flow [5,6]. The heat transfer from the top to the bottom of the weld decreases with increasing plate thickness. The temperature plays a dominant role in the plastic flow of the material [7]. Therefore, the temperature and material flow gradients along the thickness of the weld are severe in thick plates [8,9]. Silva-Magalhães et al. measured the welding temperature of 20 mm thickness AA6082 using thermocouples and the TWT method and found that the temperature difference between the top and bottom of the weld was 82 °C [10]. The obvious temperature gradient in thick plate joints resulted in microstructural variation [11]. Yang et al. reported the existence of a grain size gradient in the welding nugget zone (WNZ) due to the great temperature and strain gradient along the thickness direction [12]. The gradient characteristics of thick plates not only affect the microstructure and properties, but also greatly influence the welding formation and process window. Wang et al. believed that achieving high weld quality across the weld thickness was challenging due to the large differences in flow stress at different heights [13]. Tewari et al. studied the effect of process parameters on the tensile strength of AA1350 thick joints using the Taguchi experimental design and obtained the optimal welding speed of 80 mm/min [14]. The low welding speed produces a large heat input, causing serious joint thermal damage [15,16]. Accelerating welding speeds to minimize heat input can effectively enhance joint performance and increase production efficiency in industries [17,18]. However, increasing welding speeds further reduces heat input and material flow at the bottom of the weld, where defects are prone to occur [19]. Therefore, it is still a challenge to increase the welding speed for the welding of thick plates.

Although much research has been conducted to illustrate the phenomenon of temperature and material flow gradients in thick plate welds and to investigate the variation in microstructure and properties, it is crucial to study the effects of temperature and material flow gradients on the welding efficiency for thick plate FSW. Heat input, material flow, and mechanical properties are under the influence of the tool design [20,21,22,23]. However, there are some differences in the role of pins in thick and thin plates. Some scholars have demonstrated that the conical structure promotes the downward flow of plasticized material, and the increased conical angle contributes to a more uniform temperature distribution along the weld thickness [24]. However, the large conical angle reduces the pin contact surface at the bottom of the weld, possibly further reducing heat input and material flow at the bottom of the thick plate. Therefore, it is important to study the gradient generated by different pins and the influence of gradient on the weld formation for widening the welding process window of the thick plates.

In this study, a shoulder with a groove structure and a small penetration depth was used to achieve the micro-weld thinning. The characteristics of weld formation, microstructure, and mechanical properties under different temperatures and material flow gradients along the weld thickness were studied by two pins. The influence of the gradient on the welding process window is also discussed, which is expected to improve the welding efficiency and quality of FSW for medium-thick plates.

## 2. Materials and Methods

AA6082-T6 alloy is regarded as a desirable material for high-speed train body structure due to its excellent corrosion resistance and high specific strength. AA6082-T6 plates, 10-mm-thick, were selected as the base metal whose tensile strength and elongation were 310 MPa and 13.3%, respectively. The chemical compositions of the alloy are listed in Table 1. The welding tool was made of H13 steel. The shoulder had a diameter of 20 mm and two grooves with a depth of 0.5 mm that could store plasticized materials to avoid overflowing. Two profiles of conical threaded pins were used to weld plates. The conventional conical pin with a root diameter of 10 mm, tip diameter of 4.7 mm, and a length of 9.8 mm was named Pin-1. The large conical pin with a root diameter of 10 mm, tip diameter of 6.5 mm, and a length of 9.8 mm was named Pin-2. Both pins had triple milling facets to enhance material flow along the circumference of the pin. The schematic of the micro-weld-thinning welding process is shown in Figure 1. According to our previous work, the shoulder penetration depth, tilt angle, and rotational velocity were 0.1 mm, 1.5°, and 400 rpm, respectively. The influence of welding speeds on joints was mainly investigated, which were 100 mm/min to 500 mm/min. Pin-1 at a rotational velocity of 400 rpm and a welding speed of 100 mm/min is defined as Pin-1-400-100, and descriptions under other welding parameters are similar.

The surface morphologies of joints were observed using a digital camera. The three-dimensional morphologies were characterized by an optical microscope (OM, Keyence VHX-1000E, Osaka, Japan). Microstructural and mechanical samples were cut perpendicular to the welding direction in the center of the weld. The cross-sections of joints were polished and etched by Keller’s reagent, followed by OM to characterize the metallography morphologies. The microstructure evolution of joints was observed using the scanning electron microscope (SEM, HITACHI SU5000, Tokyo, Japan). The temperature evolution during the welding process was detected by K-type thermocouples (OMEGA HH-K-24, Hartford, CT, USA). Figure 2b shows the positions of blind holes with a diameter of 1.5 mm for presetting K-type thermocouples. A 0.3 mm thick pure copper foil between the plates was used as marker material to investigate material flow patterns during the welding process.

The microhardness maps were measured using a microhardness tester (HXD-1000TM, Shanghai, China) with a testing load of 200 g and a dwelling time of 10 s. The distance between two adjacent indentations in each row was 1 mm. The tensile properties were tested at room temperature using an tensile tester (Instron 5569, Boston, MA, USA) with a crosshead speed of 2 mm/min. The dimension of tensile samples of the whole joint was 145 mm × 12.5 mm × 10 mm, according to GB/T 228.1-2010. The tensile samples were sliced into three equal layers with a thickness of 3.2 mm to analyze the tensile properties of different regions, the detailed geometry of which can be found in Figure 2c. The tensile fracture features were observed using SEM.

## 3. Results and Discussion

### 3.1. Surface Integrity

The surface morphologies of the joints by the two pins are presented in Figure 3 and Figure 4. Both tools are capable of producing joints with smooth surfaces and no defects. The joint at a welding speed of 100 mm/min shows slight flashes on both sides of the weld. It can be seen from the three-dimensional surface morphology that the height of the flash gradually decreases with the increase in the welding speed. This is due to the higher heat input generated by the low welding speed resulting in sufficient material softening [25], which promotes the flow of plasticized material out of the shoulder. Increasing the welding speed reduces the heat input, thereby suppressing the generation of flashes. In addition, the welds under Pin-2 have larger flash at low welding speeds because the increase in pin diameter further increases the heat input and aggravates material softening. Figure 5 illustrates the statistical value of weld thinning. Weld thinning is calculated by the height difference between the weld and the base metal (BM). The maximum weld thinning is 0.08 mm at a welding speed of 100 mm/min, which is still less than 1% of the BM thickness. When welding speeds are faster than 200 mm/min, weld thinning is significantly reduced, all of which are less than 0.04 mm. The weld thinning is 0.02 mm at a welding speed of 400 mm/min, which is only 0.2% of the BM thickness. When the welding speed is further increased to 500 mm/min, weld thinning increases slightly due to the excessively fast welding speed pushing the plasticized material to both sides of the weld. The thermo-plasticized materials distanced away from the pin flow upward during the FSW process. Those materials are forged by the rotating shoulder, but a few materials flow outward along the rotating shoulder to form flashes [26]. The groove on the shoulder can store more plasticized materials, effectively preventing materials from flowing out of the shoulders. Therefore, the shoulder with grooves and small plunge depths can achieve micro-weld-thinning joints for medium-thick plates.

### 3.2. Thermal Analysis

Figure 6 shows temperature curves at different positions of the joints by Pin-1 and Pin-2, respectively. The peak temperature is related to the process parameters and tool structure. A gradual decrease in peak temperature with increasing welding speed has been demonstrated in the literature [27,28]. Therefore, comparing the effect of the two pins on the temperature distribution is the focus here. The peak temperatures measured at the top and bottom of the joint by Pin-1 are 358.5 °C and 321.5 °C, respectively, with a temperature difference of 37 °C. The top of the joints experiences much more frictional heat due to the friction of the shoulder, while the bottom of the joints experiences less heat due to the small-sized tip of the pin. Therefore, the temperature distribution exhibits a significant gradient. The peak temperature at the top of the joint by Pin-2 is consistent with that of Pin-1 because both welding tools have the same shoulder size. The peak temperature at the bottom of the joint by Pin-2 increases to 347.6 °C. The temperature difference between the top and bottom is 12 °C, which is 67.5% lower than the joint by Pin-1. The larger size tip of Pin-2 increases the frictional area and generates more frictional heat, which can be seen from the relationship between the heat generated model and the pin radius [29,30].

### 3.3. Macro and Microstructures

Figure 7 and Figure 8 show macrographs in cross-sections of the joints produced by the two pins at various welding speeds, respectively. The WNZ exhibits a typical bowl shape with a slight asymmetry. The joint of Pin-1 is defect-free at a welding speed of 100 mm/min, suggesting adequate material flow throughout the welding process. Small cavities and tunneling flaws appear at the bottom weld on the AS when the welding speed reaches 200 mm/min, as seen in the high magnification image in the red rectangle on the right. The size of the flaws formed at the bottom weld gradually increases as the welding speeds increase. The peak temperature decreases with increasing welding speeds. The materials at the bottom weld have a greater resistance to flow due to the reduced heat input [31,32]. Therefore, plasticized materials are difficult to flow sufficiently from the RS to the AS, resulting in cavities and tunneling flaws. The joints of Pin-2 remained defect-free when welding speeds were increased to 500 mm/min. Increased tip diameter increases heat input at the bottom weld, which improves the bottom formation. The welding speed is allowed to increase by 500% by Pin-2, significantly broadening the welding window of micro-weld-thinning FSW for medium-thick plates.

Figure 9 depicts the pin diameter and the WNZ width at 200 mm/min welding speed. The WNZ width decreases sharply at first and then linearly with increasing distance from the weld surface. The width of the WNZ at the top sections is similar for both pins, which is 4 mm large than the diameter of the pin. The top layer materials experience substantial heat input from the same shoulders and have a strong flow behavior under the combined shearing action by pins and shoulders. The heat input gradually decreases with increasing distance from the surface. The bottom plasticized materials flow mainly under the shearing by pins, so there are fewer deformed materials around the pin. Figure 9b shows the distribution of dimensional differences between the WNZ and the pin. The difference values under Pin-1 decrease significantly as the distance from the surface increases and eventually reach near zero at the bottom. The difference values under Pin-2 are much higher than those under Pin-1, measuring around 0.6 mm at the bottom. The large diameter pin increases heat and material deformation, resulting in more plasticized materials around the pin, which is beneficial for materials to fill the hole left by the traveling pin.

The distribution of deformed copper foils in the joints by Pin-1 and Pin-2 were compared to understand the effect of pin profiles on material flow behavior. Figure 10 shows the morphological distribution of deformed copper foil fragments in the joint at 200 mm/min welding speed. The copper foil fragments are deposited mainly in the WNZ near the AS side. The original copper foil is broken under the shearing and extruding by the pin and flows from the RS to the AS under the centrifugal force of the rotating pin, filling the cavity behind the pin. The copper fragments by Pin-1 exhibit larger size and aggregation and are less distributed at the edge of the WNZ (Figure 10a). However, the degree of copper foil fragmentation increases in the joint by Pin-2, resulting in small fragments. Numerous white fragment bands are deposited near the edge of the WNZ and are distributed continuously along the thickness direction (Figure 10c). It shows plasticized materials have good flow behavior in horizontal and thickness directions. Metallograph observation of the horizontal cross-section at 0.5 mm from the bottom surface to better investigate material flow behavior in the horizontal direction. The copper fragments in the joint by Pin-1 are short and coarse (about 0.08 mm in width and 0.65 mm in length) and are distributed fairly symmetrically on both sides of the center line. The copper fragments by Pin-2 are thin and long with an arc morphology (about 0.05 mm in width and 1.95 mm in length) and are deposited mainly on the side of AS. The width of the depositional zone on the left side of the center line is about 3.2 mm, while on the right side is only 1.3 mm. The enlarged tip pin increases the friction area at the bottom joint to provide more heat, which promotes stronger flow for the bottom materials. Therefore, more plasticized materials move towards AS under the Pin-2, completely filling the cavity behind the pin. The good formation at the bottom improves the mechanical properties of joints and facilitates the broadening of the process window.

The morphology of precipitates ((Fe,Mn)_3_SiAl_12_ and Mg_2_Si) from aluminum alloys changes due to heat and material deformation during welding [5]. Figure 11 depicts the distribution of precipitates at the bottom joint welded by two pins. The precipitates are distributed finely and uniformly in the WNZ. The WNZ experienced severe plastic deformation, resulting in the precipitates being broken up into small particles due to mechanical stirring by the pin. The precipitates are smaller in size under Pin-2 than that under Pin-1. Numerous broken zones are present around the larger particles, as shown by the red circle in Figure 11c. The high magnification images reveal that the precipitates under Pin-1 are larger, ranging in size from 4 to 7 μm, and no small particles distribute around the larger precipitates (Figure 11b). The size of precipitates decreases dramatically under Pin-2, mainly in the range of 1–4 μm. Many small precipitates distribute around the larger particles (less than 1 μm in size, as indicated by arrows in Figure 11d). It indicates that the precipitates at the bottom weld are severely broken under Pin-2. The larger diameter of pins promotes material flow at the bottom and enhances the degree of plastic deformation.

### 3.4. Mechanical Properties

#### 3.4.1. Microhardness

The microhardness maps in cross-sections of typical joints are presented in Figure 12. The microhardness of various weld zones is lower than that of BM due to the dissolution and coarsening of precipitates during the thermal cycle [33]. The lowest microhardness zone of the joint by Pin-1 is almost within the range of the stirring pin. In addition, the microhardness decrease is most pronounced at the bottom WNZ, particularly near the side of AS. The reduced heat input at the bottom weld leads to the generation of cavities. In addition, low peak temperature inhibits the re-precipitation of precipitates, thus reducing the microhardness [34]. The lowest microhardness zone of the joint by Pin-2 is on the outside of the pin. It is within the heat-affected zone (HAZ) near the side of BM for the top and middle layers and within the HAZ near the side of WNZ for the bottom layer. In addition, the softened zone narrows with increasing welding speeds, and the lowest microhardness zone becomes closer to the WNZ. Figure 13 shows the profiles of microhardness distribution in the bottom layer. The microhardness by Pin-2 exhibits typical “W” shapes, while that by Pin-1 exhibits “U” shapes. The microhardness of the WNZ by Pin-2 is significantly larger than that by Pin-1 under the same welding parameters. Increased pin diameter significantly increases heat input and improves the flow of plasticized materials at the bottom, which successfully eliminates the cavities. The high peak temperature promotes the re-precipitation of precipitates, thereby increasing the microhardness. In addition, the minimum microhardness values increase slightly with increasing welding speeds due to the reduced thermal cycle alleviating the thermal damage of materials.

#### 3.4.2. Tensile Properties

Figure 14 illustrates the tensile properties of the joints by the two pins at various welding speeds. For the joint by Pin-1, the tensile strength increases first and then decreases as welding speeds increase, while the elongation gradually decreases. The ultimate tensile strength and elongation reach 238 MPa and 4.8% at a welding speed of 200 mm/min, equivalent to 77% and 36% of BM, respectively. At a welding speed of 100 mm/min, the tensile strength decreases slightly while the elongation increases. High heat input causes the dissolution and coarsening of precipitates but promotes material flow to limit the formation of defects. The reduction in heat input at higher welding speeds reduces the loss of strength, while insufficient material flow at the bottom joint leads to forming small-sized cavities that reduce elongation. A significant decrease in tensile strength and elongation occurs at welding speeds above 300 mm/min. The further reduced heat input increases defects, severely degrading the tensile properties. For the joint by Pin-2, the tensile strength and elongation gradually increase with increasing welding speeds because the shortened thermal cycle inhibits the dissolution and coarsening of precipitates. Tensile properties stay nearly at the same high level at welding speeds above 300 mm/min. The ultimate tensile strength and elongation reach 247 MPa and 8.7% at a welding speed of 500 mm/min, equivalent to 80% and 65% of BM, respectively.

Figure 15 illustrates the tensile properties of different layers of typical joints. The ultimate tensile strength of the top, middle, and bottom layers of the joint by Pin-1 are 237 ± 5 MPa, 234 ± 3 MPa, and 220 ± 6 MPa, respectively. The bottom layer is a weak zone for the joint. Lower heat input and poor material flow occur at the bottom due to the small diameter of the tip pin, resulting in defects. The ultimate tensile strength of the top, middle, and bottom layers of the joint by Pin-2 are 236 ± 4 MPa, 241 ± 3 MPa, and 238 ± 1 MPa, respectively. The tensile strength of each layer is equivalent, and the bottom layer, in particular, does not reduce. Increasing the diameter of the tip pin improves the joint formation at the bottom, resulting in uniform loading. The tensile properties of each layer remain consistent when welding speed increases to 400 mm/min. Clearly, the joint by Pin-2 has good mechanical properties under a wide welding process window, indicating the potential to improve productivity.

Figure 16 and Figure 17 display the tensile fracture features of the joints by the two pins at different welding speeds, respectively. For the joint by Pin-1, the bottom fracture location is near the edge of the WNZ at a welding speed of 100 mm/min. When the welding speed is 200 mm/min, the bottom fracture path is at the WNZ with significant angular deflection and a height of 2.4 mm. The cavities in the bottom WNZ act as crack sources, promoting the initiation and propagation of cracks. The fracture zone located in the WNZ gradually increases with increasing welding speeds. It reaches a height of 4.3 mm at a welding speed of 400 mm/min, nearly half the thickness of the joint. These results are consistent with the defect morphology shown in Figure 7. All the joints by Pin-1 are fractured on the AS side due to the small tip pin leading to poor material flow at the bottom joint, which reduces the metallurgical bonding at the edge of the WNZ. For the joint by Pin-2, all specimens fracture along a path showing a 45° angle to the tensile axis, a typical feature of shear fracture patterns. The bottom fracture location approaches from HAZ to WNZ with increasing welding speeds. It is completely inside the HAZ at a welding speed of 100 mm/min, and occurs at the outer edge of the WNZ at a welding speed of 200 mm/min and at the inner edge of the WNZ when the welding speed is 500 mm/min. The heat generation at the bottom gradually decreases with increasing welding speeds, suppressing thermal damage in the HAZ. The metallurgical bonding of the boundary between HAZ and WNZ deteriorates with decreasing heat input, thus changing the fracture location from HAZ to WNZ.

The SEM images of fracture surfaces at the bottom of typical joints are shown in Figure 18 to further clarify the fracture features. Larger cavities are found in the joint by Pin-1. The fracture morphology between two adjacent cavities exhibits numerous tiny dimples and tearing edges, as well as broken precipitates at the bottom of the dimples. The bottom fracture location occurs in the WNZ as a result of the generation of defects, as shown in Figure 16. Cracks are generated through the cavities and propagate to adjacent cavities under tensile loading. The grains in WNZ achieve dynamic recrystallization under severe plastic deformation [35], so the equiaxed grains lead to fracture morphology exhibiting a large number of tiny dimples. A large number of larger-sized dimples without defects are found in the joint by Pin-2 (Figure 18c,d). The fracture is located in the HAZ at 200 mm/min welding speed, where grains and precipitates are coarsened under higher heat input. These precipitates become the nuclei of the cracks, reducing the strength of the joint. The fracture surface is dominated by fine and deep dimples as welding speeds increase to 400 mm/min. The fracture is located in the TMAZ, where the grains and precipitates are significantly smaller than those in the HAZ. The smaller precipitates promote the uniform growth of micro voids and eventually the formation of numerous small dimples, thus enhancing the tensile strength and elongation of joints.

## 4. Conclusions

A micro-weld-thinning FSW was proposed to join medium-thick plates. The effects of temperature and material flow gradients along the weld thickness on weld formation, microstructure, and mechanical properties were studied. Based on the investigated results, the following conclusions can be extracted:The shoulder with a groove and a small plunge depth effectively prevented the overflow of plasticized materials. The weld thinning of all joints maintained a low level, with a minimum of 0.02 mm, which was 0.2% of the BM thickness;The low peak temperature and weak material flow at the bottom weld caused by the small-tip pin were not favorable to the flow of plasticized materials to AS, which led to the susceptibility to defects at the bottom of the weld, as well as low microhardness and strength;The large-tip pin increased heat input and material flow at the bottom weld. The temperature difference between the top and bottom was reduced by 67.5% compared to the small-tip pin. The material flow was enhanced in both horizontal and thickness directions, effectively suppressing the generation of root defects;The large-tip pin increased the welding speed of defect-free joints from 100 mm/min to 500 mm/min, which significantly increased productivity. The ultimate tensile strength and elongation of the joint reached 247 MPa and 8.7%, respectively, equivalent to 80% and 65% of the BM.

## Figures and Tables

**Figure 1 materials-15-06579-f001:**
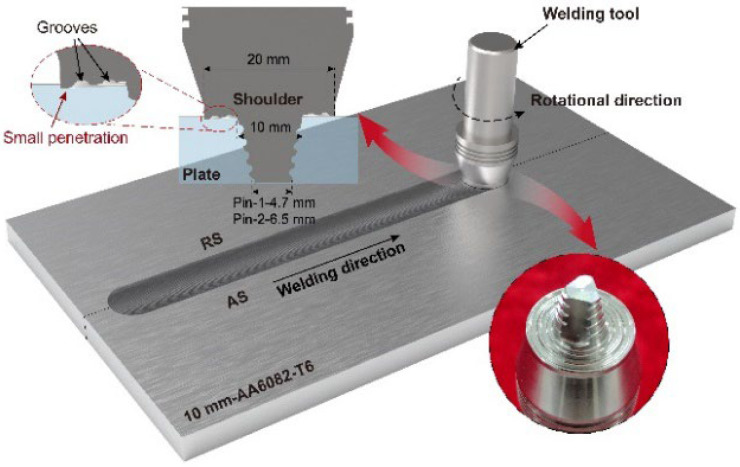
Schematic of micro-weld thinning FSW process.

**Figure 2 materials-15-06579-f002:**
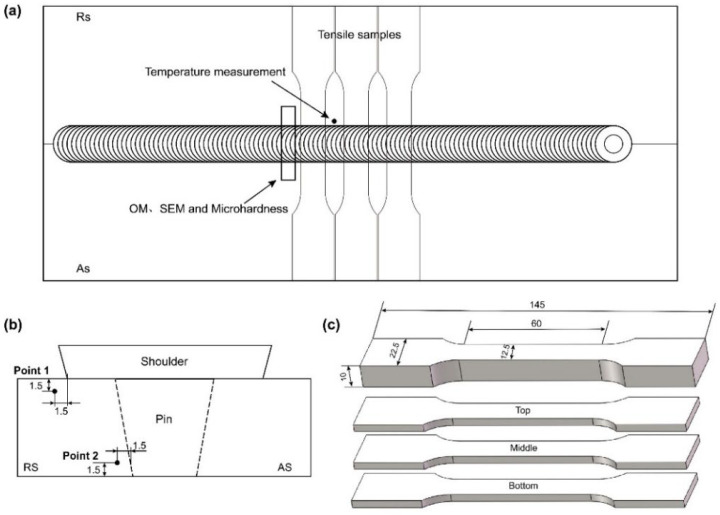
(**a**) Measurement position, (**b**) temperature measurement points, and (**c**) tensile specimens (mm).

**Figure 3 materials-15-06579-f003:**
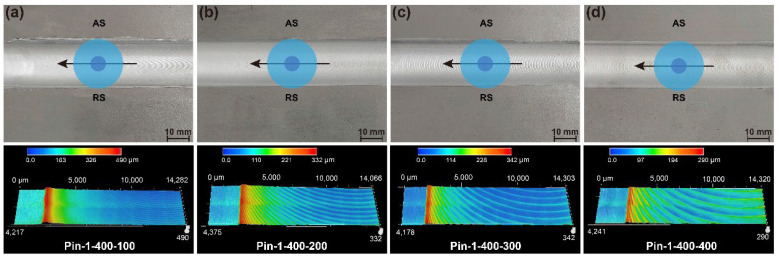
Surface morphologies (**a**) Pin-1-400-100, (**b**) Pin-1-400-200, (**c**) Pin-1-400-300, and (**d**) Pin-1-400-400.

**Figure 4 materials-15-06579-f004:**
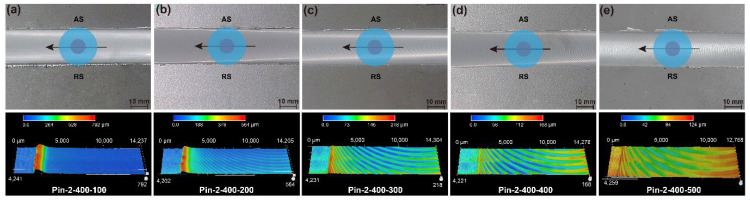
Surface morphologies (**a**) Pin-2-400-100, (**b**) Pin-2-400-200, (**c**) Pin-2-400-300, (**d**) Pin-2-400-400, and (**e**) Pin-2-400-500.

**Figure 5 materials-15-06579-f005:**
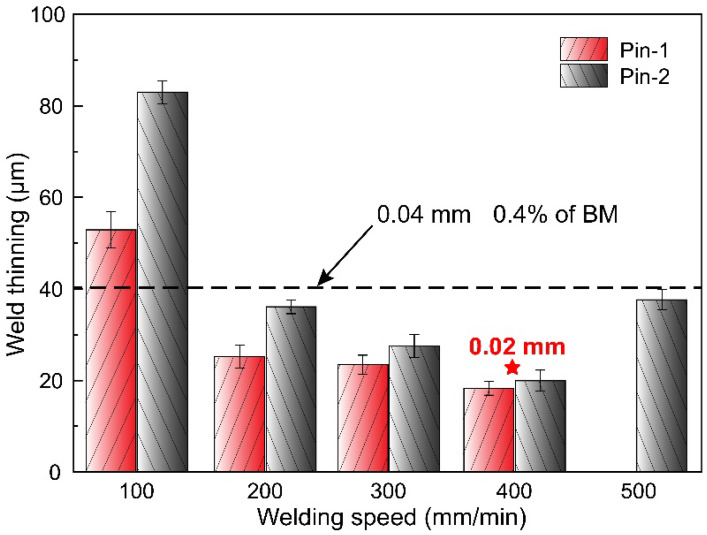
Weld thinning of joints at different welding speeds.

**Figure 6 materials-15-06579-f006:**
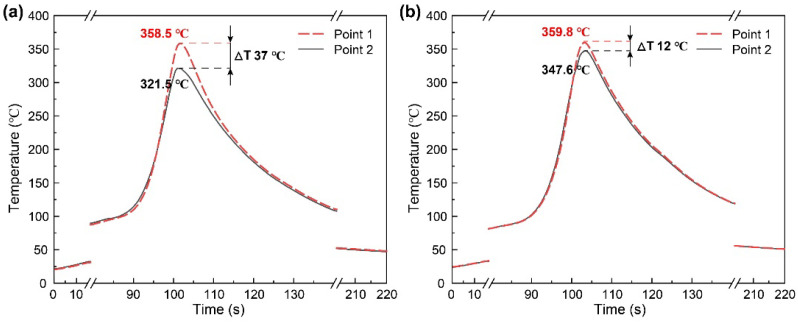
Temperature curves at different positions of (**a**) Pin-1-400-200 and (**b**) Pin-2-400-200.

**Figure 7 materials-15-06579-f007:**
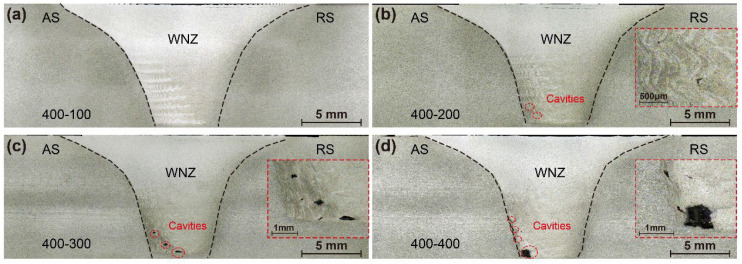
Macrostructure of joints by Pin-1 (**a**) 100 mm/min, (**b**) 200 mm/min, (**c**) 300 mm/min, and (**d**) 400 mm/min.

**Figure 8 materials-15-06579-f008:**
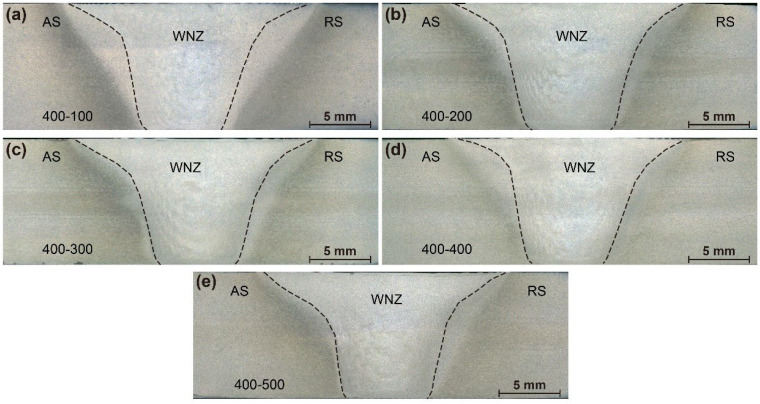
Macrostructure of joints by Pin-2 (**a**) 100 mm/min, (**b**) 200 mm/min, (**c**) 300 mm/min, (**d**) 400 mm/min, and (**e**) 500 mm/min.

**Figure 9 materials-15-06579-f009:**
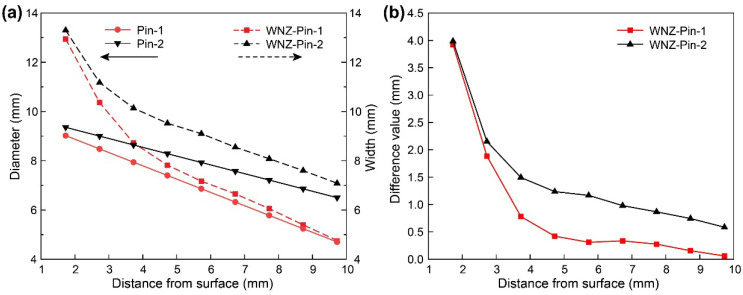
(**a**) Pin diameter and WNZ width and (**b**) difference value between pins and WNZ.

**Figure 10 materials-15-06579-f010:**
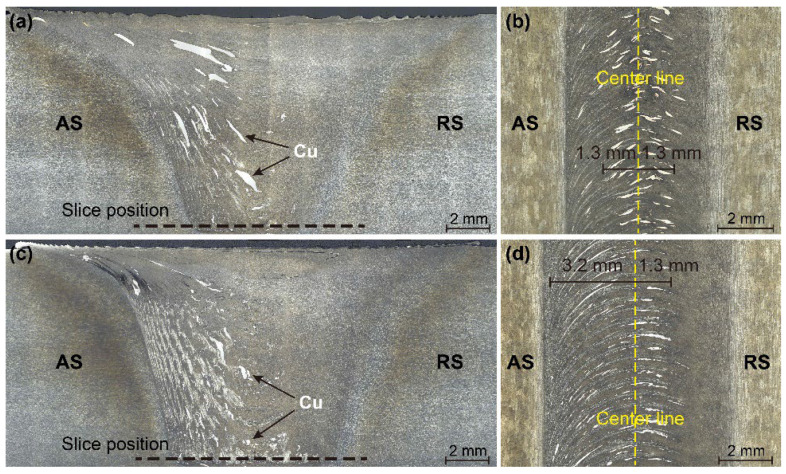
Distribution of deformed copper foil in joints (**a**,**b**) Pin-1 and (**c**,**d**) Pin-2.

**Figure 11 materials-15-06579-f011:**
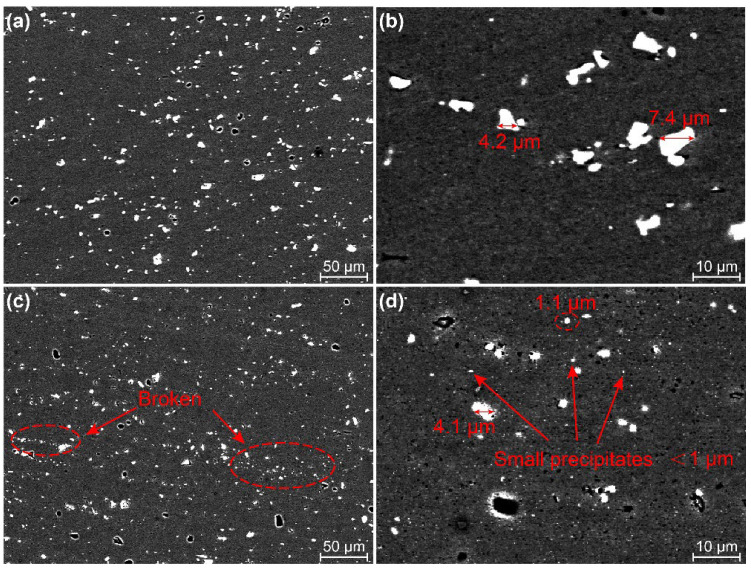
Distribution of precipitates at the bottom WNZ (**a**,**b**) Pin-1 and (**c**,**d**) Pin-2.

**Figure 12 materials-15-06579-f012:**
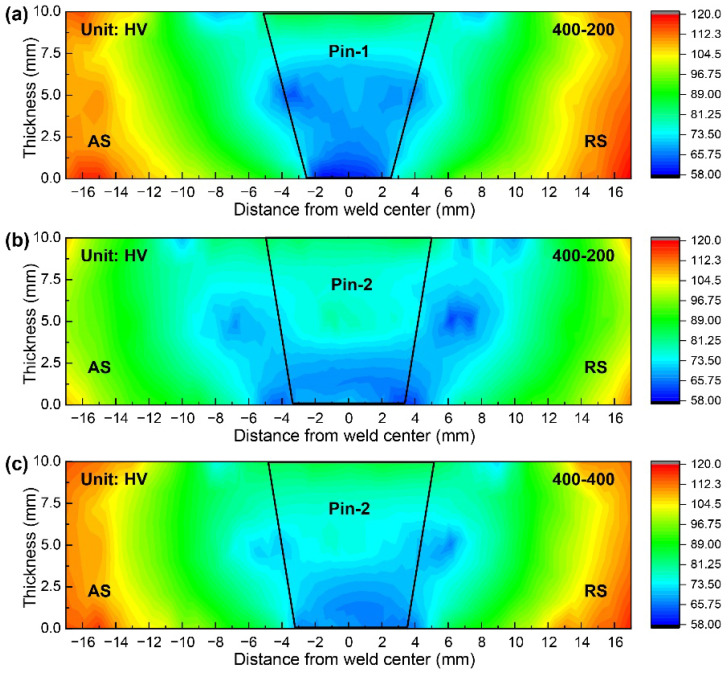
Microhardness maps of typical joints (**a**) Pin-1-400-200, (**b**) Pin-2-400-200, and (**c**) Pin-2-400-400.

**Figure 13 materials-15-06579-f013:**
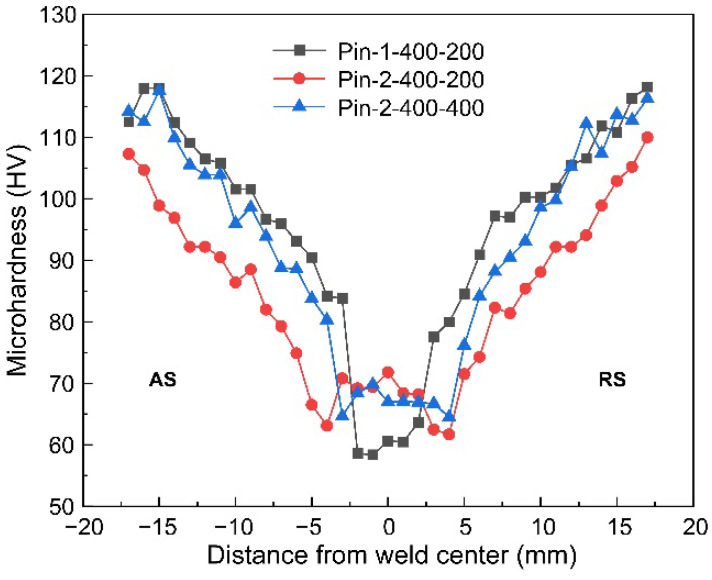
Microhardness distributions of bottom layers.

**Figure 14 materials-15-06579-f014:**
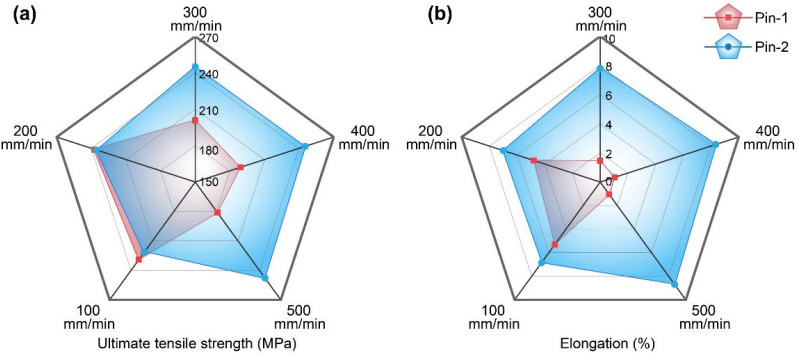
(**a**) Ultimate tensile strength and (**b**) elongation.

**Figure 15 materials-15-06579-f015:**
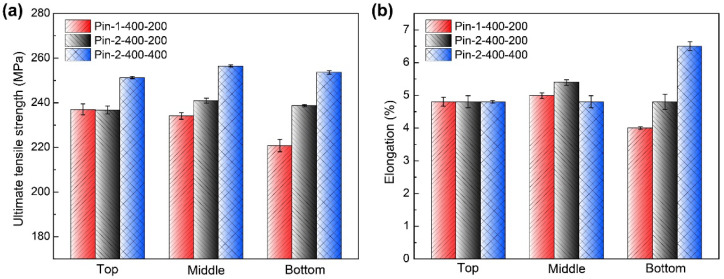
Tensile properties of different layers (**a**) ultimate tensile strength and (**b**) elongation.

**Figure 16 materials-15-06579-f016:**
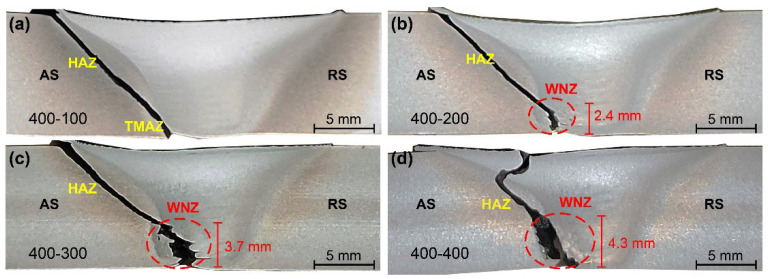
Fracture locations of joints by Pin-1 (**a**) 400-100, (**b**) 400-200, (**c**) 400-300, and (**d**) 400-400.

**Figure 17 materials-15-06579-f017:**
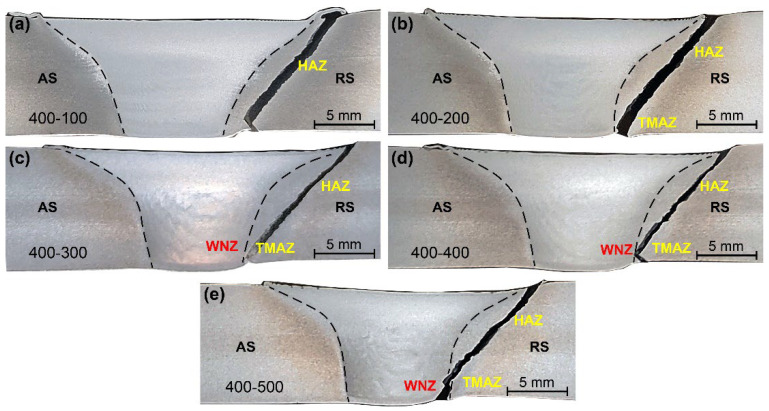
Fracture locations of joints by Pin-2 (**a**) 400-100, (**b**) 400-200, (**c**) 400-300, (**d**) 400-400, and (**e**) 400-500.

**Figure 18 materials-15-06579-f018:**
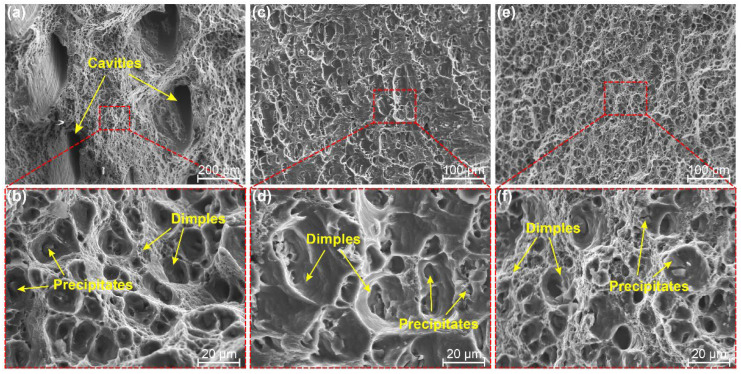
Fracture morphologies at the bottom joints (**a**,**b**) Pin-1-400-200, (**c**,**d**) Pin-2-400-200, and (**e**,**f**) Pin-2-400-400.

**Table 1 materials-15-06579-t001:** Chemical compositions of AA6082-T6 (wt%).

Mg	Zn	Si	Mn	Fe	Cr	Cu	Ti	Al
1.10	0.31	0.81	0.15	0.75	0.07	0.23	0.19	Bal.

## Data Availability

Not applicable.

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
