# Peer review of "Effect of Temperature and Material Flow Gradients on Mechanical Performances of Friction Stir Welded AA6082-T6 Joints"

_materials, 2022, doi:10.3390/ma15196579_

Round 1

Reviewer 1 Report

This review paper deals with the ¨ Effect of temperature and material flow gradients on mechanical performances of friction stir welded AA6082-T6 joints¨. The manuscript topic is exciting but needs major revision:

1- The writing of English needs significant revision. Some typos and grammatical errors can be seen in the manuscript.

2- The introduction needs to be improved. The author should focus on the FSW of thick plates. The presented literature is not related to the manuscript scope. There are available papers that focus on the FSW of thick plates.

3- The benefits and drawbacks of this review are not clear. The authors aimed for what they wanted to present, but a piece’s literature was missed and needs to add to the introduction section. The problem that the authors want to solve is not apparent.

4- Please add the manufacturer, company, and model of used thermocouples.

5- please discuss more relation pins and tool traveling speed with formed flash in Fig. 4.

6- Please discuss more related tool traveling speed with recorded heat in section 3.2.

7- Please unify all results according to the tool traveling speed. In some sections, the authors reported 500mm/min; in a couple of them, this traveling speed result was removed. 

Reviewer 2 Report

The paper describes the temperature effects of  joining of aluminium plate by friction stir welding using different tool geometry . The evaluation of the results is finally carried out by non-destructive and destructive test methods (tensile, microhardness , microstructure analytics by light-microscope and SEM. From the reviewer’s point of view, the article requires a minor correction

-        Abstract must clarify and specialize the main contribution of research with outcomes

-        Dimensions of tool should be present in schematic figure

-        What the reasons that authors select two different tool geometry?

-        You should describe in detail the position of the measurement position. A sketch would be helpful.

-        Heated effected Zone must include TMAZ also in hardness as well

-        In page 5 . line 163-156 , (The larger size tip of Pin-2 increases the frictional area and generates more 163 frictional heat due to the heat generation being positively correlated with the square of 164 the pin radius) explain more with References

-        the state of the introduction need add shortly extended by summarization articles which considering the recent studies  of friction welding  which are related to this manuscript specially of effect of heat input as:

-        Soori, M., et al., "Recent Development in Friction Stir Welding Process: A Review," SAE International Journal of Materials and Manufacturing. 14(1):2021

-        Nasir, Tauqir, et al. "The experimental study of CFRP interlayer of dissimilar joint AA7075-T651/Ti-6Al-4V alloys by friction stir spot welding on mechanical and microstructural properties." Nanotechnology Reviews 10.1 (2021): 401-413

Reviewer 3 Report

The authors carried out the studies on “Effect of temperature and material flow gradients on mechanical performances of friction stir welded AA6082-T6 joints”. The authors investigated the present study is systematic manner. The results are presented in the good form. The article may be considered for publication with addressing the following comments. 

 The Introduction Section looks like the general theory about the Friction Stir welding, authors are not reviewed the literature and finding the gaps to the present study. 

Authors also not provided the information why, AA 6082-T6 is chosen for the present study. 

Table 2 can be removed in the manuscript; the data is already available in line no 81 and 82. 

Line no 160, spelling mistake same side shoulder need to be replaced same size shoulder

Authors may include the XRD analysis to identify the phase formation in the present investigation

Round 2

Reviewer 1 Report

The authors addressed all comments correctly.